Scavenging vs hunting affects behavioral traits of an opportunistic carnivore

Parsons Mitchell A. 1 2
Garcia Andrew 2
Young Julie K. julie.young@usu.edu 1 2
1 Department of Wildland Resources, Utah State University , Logan , UT , United States of America
2 National Wildlife Research Center - Predator Research Facility, USDA , Millville , UT , USA
Vonk Jennifer
Electronic publication date: 2022 May 2
Publication date: 2022
Volume: 10
Electronic Location ID: e13366
Received 2022 Jan 20; Accepted 2022 Apr 11
Copyright: ©2022 Parsons et al.
Copyright year: 2022
Copyright holder: Parsons et al.
License: This is an open access article distributed under the terms of the Creative Commons Attribution License, which permits unrestricted use, distribution, reproduction and adaptation in any medium and for any purpose provided that it is properly attributed. For attribution, the original author(s), title, publication source (PeerJ) and either DOI or URL of the article must be cited.
License URL: https://creativecommons.org/licenses/by/4.0/

Keywords: Anthropogenic food, Foraging ecology, Novel object, Puzzle box, Predator-prey, Canis latrans

Funding: The U.S. Department of Agriculture National Wildlife Research Center (NWRC) This project was funded by the U.S. Department of Agriculture, National Wildlife Research Center (NWRC). The funders had no role in study design, data collection and analysis, decision to publish, or preparation of the manuscript.

==============================
Background

Human-induced changes to ecosystems transform the availability of resources to predators, including altering prey populations and increasing access to anthropogenic foods. Opportunistic predators are likely to respond to altered food resources by changing the proportion of food they hunt versus scavenge. These shifts in foraging behavior will affect species interactions through multiple pathways, including by changing other aspects of predator behavior such as boldness, innovation, and social structure.

Methods

To understand how foraging behavior impacts predator behavior, we conducted a controlled experiment to simulate hunting by introducing a prey model to captive coyotes (Canis latrans) and compared their behavior to coyotes that continued to scavenge over one year. We used focal observations to construct behavioral budgets, and conducted novel object, puzzle box, and conspecific tests to evaluate boldness, innovation, and response to conspecifics.

Results

We documented increased time spent resting by hunting coyotes paired with decreased time spent active. Hunting coyotes increased boldness and persistence but there were no changes in innovation. Our results illustrate how foraging behavior can impact other aspects of behavior, with potential ecological consequences to predator ecology, predator-prey dynamics, and human-wildlife conflict; however, the captive nature of our study limits specific conclusions related to wild predators. We conclude that human-induced behavioral changes could have cascading ecological implications that are not fully understood.

Introduction

Predators are strongly impacted by human-induced global change due to their large spatial and resource needs (Ripple et al., 2014). Through large-scale ecological changes, humans have altered the availability of both natural prey and non-prey resources for predators (Coon et al., 2019; Ciucci et al., 2020; Mills & Harris, 2020). Despite a frequent emphasis on the effects of the presence or absence of predator on prey (Salo et al., 2010), minimal research has addressed how the loss of prey may impact predator ecology and behavior (Rodewald, Kearns & Shustack, 2011; Newsome et al., 2015a). Availability of prey has decreased for apex predators globally (Wolf & Ripple, 2016; Vinks et al., 2021), although prey resources have increased in some areas (Prange, Gehrt & Wiggers, 2004). Simultaneously, human activity has altered scavenging opportunities (Yirga et al., 2012; Fey et al., 2015) and increased access to anthropogenic resources (Newsome et al., 2015a; Ciucci et al., 2020). Some predators successfully hunt in disturbed environments, often focusing on small prey that become more abundant with human presence (Prange, Gehrt & Wiggers, 2004; Coon et al., 2019; Rodriguez, Lesmeister & Levi, 2021). However, other predators are likely to shift to using non-prey resources that have become increasingly abundant (Yirga et al., 2012; Newsome et al., 2015a; Cozzi et al., 2016). In some situations, predators have shifted almost exclusively to non-prey resources (Parsons, Newsome & Young, 2022). Changes to resource availability are most evident in globally expanding urban and suburban environments (McDonnell & Hahs, 2015; Robins et al., 2019). However, human influence also alters resource availability in non-urban ecosystems through livestock production, rural landfills, and spillage of food during transportation (Ciucci et al., 2020; Mourier, Claudet & Planes, 2020). These shifts in resource availability may influence predator resource use and ecological interactions (Beckmann & Berger, 2003; Ciucci et al., 2020; Gámez & Harris, 2021).

Changes in resource use could alter predator ecology through multiple pathways. First, the use of alternative resources will affect interactions with prey species and other predators (Manlick & Pauli, 2020). Second, predators incorporating more scavenged foods may alter their behavior (Chilvers & Corkeron, 2001; Cozzi et al., 2016; Parsons, Newsome & Young, 2022), and there is a potential relationship between individual diet specialization and animal personality (Toscano et al., 2016). Animal personality describes consistent differences in behaviors between individuals through time, space, and ecological contexts (Young, Mahe & Breck, 2015; Toscano et al., 2016). However, these consistent differences between individuals do not preclude changes to behavioral traits in response to the environment (Sih, Bell & Johnson, 2004).

Personality traits that are likely related to foraging behavior include boldness, persistence, and innovation (Table S1). Boldness describes an individual’s risk aversion (Brooks, Kays & Hare, 2020) and is linked to foraging decisions including willingness to forage alone and to explore novel foraging opportunities (Michelena et al., 2009; Kurvers et al., 2012; but see Carter et al., 2012). Persistence is defined as the amount of time animals spend interacting with a stimulus, or the rate of interaction, which impacts an animal’s ability to acquire resources from novel sources (Chow, Lea & Leaver, 2016; Jacobson et al., 2021). Innovation describes the ability of animals to display new behaviors or modify present behaviors to solve novel problems (Johnson-Ulrich, Johnson-Ulrich & Holekamp, 2018; Barrett, Stanton & Benson-Amram, 2019); a key trait to acquiring resources in new environments. Bold, persistent, and innovative individuals are more likely to succeed when changing foraging strategies as they will be more likely to interact with and solve new foraging problems (Sol, 2009; Sol, Lapiedra & González-Lagos, 2013). Such behavioral changes could also lead to cascading effects, including increased human-wildlife conflict (Barrett, Stanton & Benson-Amram, 2019; Brooks, Kays & Hare, 2020) and altered predator–prey dynamics (Szopa-Comley et al., 2020). Although abundant research has addressed how prey behavioral traits mediate predator–prey interactions (Blake & Gabor, 2014; Toscano, 2017), knowledge regarding the importance of predator traits is lacking (but see Parsons, Newsome & Young, 2022).

Foraging behavior may also influence social behavior of predators. Foraging behavior relates to group size in coyotes (Canis latrans; Bowen, 1981), cooperative behavior in bottlenose dolphins (Tursiops truncates; Díaz López & Shirai, 2008), and conspecific tolerance in multiple species (Peirce & Van Daele, 2006; Fallows, Gallagher & Hammerschlag, 2013; but see Gilchrist & Otali, 2002). Similarly, the abundance of easily captured food in urban environments may increase conspecific tolerance, reduce territoriality, and increase group sizes for predators exploiting these resources (Bateman & Fleming, 2012). Changes to social structure are complex and could alter the distribution of predators on the landscape as well as the landscape of fear experienced by mesopredators and prey species (Prugh et al., 2009; Kohl et al., 2019).

Opportunistic predators that integrate diverse food items into their diet are most likely to alter resource use and experience behavioral changes (Eddine et al., 2020; Parsons, Newsome & Young, 2022). In North America, coyotes are ubiquitous opportunistic carnivores that have vastly expanded their range over the past century (Hody & Kays, 2018). This expansion includes urban environments where coyotes consume both prey and non-prey resources (Fedriani, Fuller & Sauvajot, 2001; Gehrt, 2007; Murray et al., 2015) and display different behavioral traits than rural coyotes, including increased boldness and exploratory behavior (Breck et al., 2019; Brooks, Kays & Hare, 2020). Coyotes in urban environments differ widely in their use of anthropogenic foods, likely due to the broad spectrum of urban conditions (Schell et al., 2020). Anthropogenic foods constitute 50% or more of coyote diet in some locations (Fedriani, Fuller & Sauvajot, 2001; Murray et al., 2015; Newsome et al., 2015b) and reliance on anthropogenic food may increase the prevalence of human-wildlife conflicts (White & Gehrt, 2009; Schell et al., 2021). Because of their range expansion associated with human development and ability to consume diverse resources, coyotes present a valuable model system for understanding how foraging strategies impact predator behavior and subsequently influence community ecology and human-wildlife conflict.

We used captive coyotes to investigate relationships between foraging behavior and other behavioral traits. Conducting this work with captive animals allowed us to repeatedly observe the same individuals over one year, control their food availability and delivery, and maintain a relatively constant environment; all factors that are infeasible to manage in wild systems where changes to available food resources often occur with changes to the physical environment (Coon et al., 2019; Robins et al., 2019). By working with captive individuals, we were able to isolate the effect of foraging behavior from other factors. However, working with captive animals that were exclusively scavengers, contrary to their wild counterparts, meant we had to change the foraging behavior by introducing hunting opportunities. Captive animals are typically scavengers, with hunting introduced to prepare them for release into the wild (Vargas & Anderson, 1999) and as an enrichment activity (Markowitz & LaForse, 1987). The opposite shift would be expected with wild predators—from hunting to scavenging (e.g., Fedriani, Fuller & Sauvajot, 2001; Murray et al., 2015). Even so, shifts from scavenging to hunting have been observed in wild populations (Whitehead & Reeves, 2005). Identifying the extent of the relationship between foraging behavior and other behavioral traits is valuable because shifts in either direction require behavioral flexibility, a key trait to surviving in dynamic environments (Sol, 2009; Barrett, Stanton & Benson-Amram, 2019), and innovation in solving new foraging tasks to access novel prey or scavenging resources (Griffin & Guez, 2014; Daniels et al., 2019).

We conducted a controlled experiment to assess relationships between predator foraging patterns and individual behavior. We used a population of captive coyotes as a model system that behave similarly to wild coyotes (Shivik et al., 2009). We selected a subset of the captive coyotes to hunt prey—we introduced a prey model that resembled a lagomorph and required pursuit and capture to acquire food. Control coyotes were also given an immobile prey model to control for the novelty of the model. We compared behavioral budgets, innovation, persistence, and boldness between treatment and control coyotes, and modeled trends in each behavior separately with covariates for treatment, month, and interaction to account for seasonal, annual, and other changes not due to our hunting treatment. We hypothesized that hunting coyotes would decrease stereotyping behavior and increase time spent searching for food due to the increased stimulation of pursuing prey. Learning to capture the prey model also required interacting with a novel object, persistence, and problem-solving skills. Coyotes had to be willing to approach the prey model, interact with the model, and successfully knock the model over to acquire food. We expected that hunting coyotes would become bolder, more persistent, and more innovative than control coyotes across the study period. Finally, we expected that hunting coyotes would behave more aggressively towards an unknown conspecific due to the increased effort to acquire food. These hypotheses are based on previous literature (Williams et al., 1996; Kistler et al., 2009; Breck et al., 2019; Newsome, Howden & Wirsing, 2019), but also reflect the unique conditions of our study system because directions of behavioral change will depend both on initial conditions and how resource availability changes. Our results provide insights into how foraging behavior is linked with other behavioral traits. Results will improve our understanding of the role hunting plays on individual behavior, which could impact predator–prey dynamics (Szopa-Comley et al., 2020), human-wildlife conflict (Barrett, Stanton & Benson-Amram, 2019), and predator conservation (Bombieri et al., 2021) in wild populations.

Materials & Methods

Study site

We conducted this study at the United States Department of Agriculture National Wildlife Research Center’s (NWRC) Predator Research Facility in Millville, Utah, USA. This facility maintains ∼90 adult coyotes, typically housed as male–female pairs in outdoor enclosures (0.1–1.0 ha).

Study subjects

Our study was conducted on eight coyote pairs, four treatment pairs and four control pairs, ranging from 1.5 to 3.5 years old at the start of the study (details in Supplementary Information). We split coyotes equally among treatment groups based on age. Because our study took place across a year, two pairs of treatment coyotes and two pairs of control coyotes reared pups (April–June) during this study for colony management purposes. All coyotes used in this study were captive-born with minimal human intervention at the facility. Occasionally wild birds and mammals have entered the coyote enclosures and some coyotes have therefore killed live prey (S Brummer, pers. comm., 2019). We observed the coyotes involved in this study and their enclosures on a near-daily basis and monitored for any sign of predation and documented no cases by any of the study coyotes throughout the project.

Ethics statement

All research methods and procedures were approved by NWRC’s Institutional Animal Care and Use Committee (QA-3151).

Feeding treatments

Each coyote was fed 650 g of commercial mink food (Fur Breeders Agricultural Cooperative, Logan, Utah, USA) at least six days per week (i.e., 1,300 g per pair) by animal care staff who enter and scatter the food in each enclosure. We continued this feeding method for our control coyotes and for three days per week with our treatment coyotes.

For the hunting treatment, we created a rabbit prey model from a remote-controlled car (Everest Gen7 Sport RC Crawler, Redcat Racing, Phoenix, AZ, USA). We removed the car body and draped the car chassis with rabbit fur. We used foam to sculpt a rabbit head and attached this to the front of the car (Fig. S1). The prey model was approximately 43 × 18 × 19 cm, similar in size to a snowshoe hare (Lepus americanus; Reid, 2006). We fed treatment coyotes using the prey model three days per week to mimic a wild predator acquiring 50% of its food through hunting and 50% through scavenging. To feed coyotes with the prey model, we placed one portion of food on the prey model and placed the food-laden model in the enclosure. Initially, we left the prey model stationary and scattered additional food around it for coyotes to become comfortable with the prey model and associate it with food. As coyotes became more comfortable with the prey model, we began moving it in small, predictable circles, and slowly increased complexity and speed to an unpredictable path to simulate hunting. Upon capturing the prey model, coyotes were allowed to consume the food. We defined capture as knocking the prey model over so that it could no longer move. This required innovation from study subjects because they had to identify ways to access the mobile food. Then we retrieved the model and scattered the remaining food for the day throughout the enclosure. If coyotes did not successfully hunt on a given day, we removed the prey model after one hour and returned at least two hours later and fed the pair using the standard procedure for the facility. While this treatment does not fully mimic hunting in the wild, it does require pursuit and capture of the prey model, two key components of the hunting process (MacNulty, Mech & Smith, 2007). Control coyote pairs were fed with a stationary prey model three days per week to control for the presence of a novel object during feeding.

Behavioral tests

Feeding and nonfeeding focal sampling

We began this study in October 2019 by collecting baseline measures in a pretrial phase for all coyotes. We first introduced the prey model in December 2019. From January to November 2020, we conducted a suite of behavioral tests (Fig. S2).

We conducted monthly focal observations during both feeding and nonfeeding times to construct behavioral budgets for coyotes (Fig. S2). We modified the ethogram from Shivik et al. (2009; Table 1) and conducted continuous observations for 15 min for each coyote. We conducted shorter observations than Shivik et al. (2009) but the same length as other research with captive coyotes (Leary, Schultz & Young, 2021). Shorter observations facilitated sampling multiple times each month and allowed time for behavioral tests and daily animal care routines to be completed during daylight hours. Each month, we observed each coyote during four nonfeeding times, four feedings without the prey model, and four feedings with the prey model (Table S2), with two nonfeeding observations in the morning (6:00–10:00) and two nonfeeding observations in the afternoon/evening (14:00–21:00; depending on the time of year and day length). We conducted nonfeeding observations using a dedicated observation vehicle as a blind parked >50 m away (Schell et al., 2018). We video-recorded all feeding observations for later behavioral coding. Focal observations provided behavioral budget data and we used the proportion of time coyotes spent interacting with the prey model during prey-model feedings as a measure of persistence while hunting.

Table 1 Ethogram of behaviors used to construct behavioral budgets for coyotes.

Observers coded behavior based on the behavior and descriptions. Behaviors were then condensed into five broad categories for analysis to reduce the number of behaviors and potential for interobserver error. Modified from Shivik et al. (2009).

Behavior	Description	Broad category	
Resting	Coyote laying or sitting down	Resting	
Locomotion	Purposeful walking, trotting, or running with head up	Active	
Standing	Standing still with head raised	NA	
Foraging	Orientating, stalking, and searching at a slow pace with head lowered	Feeding	
Eating	Coyote eating with visible jaw movement	Feeding	
Aggressive	Teeth bared, biting, growling, chasing mate away from food.	Social	
Play	Playful behavior with mate, tail wagging, non-threatening posture	Social	
Neutral Social	Howling and other behaviors directed at mate that are neither aggressive, nor playful	Social	
Stereotyping	Repetitive movement with no apparent goal that is repeated for greater than 2 cycles.	Active	
Investigating car*	Head and gaze oriented towards prey model from a distance while standing or walking slowly.	Feeding/Car	
Interacting with car*	Behaviors directed towards prey model such as chasing, pouncing, scratching, or biting.	Feeding/Car	
Scent marking	Urinating or defecating –point event	NA	
Notes.

* These behaviors were only relevant during car feeding observations.

Novel object trials

We conducted a novel object test every three months starting in the pretrial phase to measure coyote boldness (Fig. S2 and Table S2). We placed a novel object near the center of the coyote enclosure and used a video camera to record interactions for one hour. Coyotes were in the enclosure when the novel object was placed and generally retreated to the far end of the enclosure 20–50 m away depending on enclosure size. Observation time started when the researcher closed the gate and exited the enclosure. We used a new novel object each period but kept approximate size consistent between periods; novel objects were the puzzle box during the first puzzle box trial (see below), two car tires attached to form a ball, a traffic cone, a 5-gallon water jug, and a cylindrical wire cage. During the initial presentation of the puzzle box, we did not include food so that we could use it as a novel object. All coyotes received the novel objects in the same order so that any differences between control and treatment coyotes could be attributed to longitudinal trends and not confounded by individuals receiving different objects at different timesteps (Schell, 2015). A trained observer coded videos to extract the latency to approach within 5 m, 1 m, and to touch the novel object.

Multi-access puzzle box tests

We used a multi-access puzzle box to test persistence and repeated innovation every three months starting in the pretrial phase (Fig. S2 and Table S2) (Auersperg et al., 2011; Johnson-Ulrich et al., 2021). The puzzle box was a 45.7 × 45.7 × 45.7 cm cube constructed with three sides of clear PVC sheeting with a door in each and three sides of white PVC sheeting (Fig. S3). One door pushed inwards, one door pulled outwards with a racquetball handle, and the third door swung open if a wooden peg was removed. During each testing period, we presented the coyotes with the puzzle box for 10, two-hour trials. The puzzle box remained in the enclosure throughout each testing period (i.e., between daily trials) to allow coyotes to become and remain familiar with its presence. We introduced the puzzle box with food inside and left all doors open for three days for the coyotes to gain familiarity with the puzzle box before trials began. On the first day of testing, we replaced any remaining food, closed all doors, and filmed interactions with the puzzle box continuously for two hours. We placed small bits of food around the doors and smeared peanut butter on the doors to encourage interaction. At the end of each two-hour test trial, we opened the doors and removed the food to prevent coyotes from learning that they could access the food by waiting for the trial to end. We left the box, with the doors open, to reduce neophobia at the beginning of subsequent days. We repeated this procedure for five consecutive days, then left the box inside the enclosure and open with food inside to give study subjects and observers a reprieve from trials for two days before repeating the test procedure on days 8–12 for a total of 10 test days each period. If a coyote solved one solution to the puzzle box, that solution was locked for the remainder of that test period to encourage interaction with other solutions (Jacobson et al., 2021). During the next trial period, all coyotes started with all solutions available to document whether coyotes used familiar solutions or novel solutions. We did not require an individual to use a solution multiple times before locking the solution (e.g., Johnson-Ulrich, Johnson-Ulrich & Holekamp, 2018) because the lack of interaction with the puzzle box (see Results) made this approach untenable. A trained observer coded puzzle box videos for the proportion of time that coyotes spent within 5 m, 1 m, and interacting with the puzzle box. We defined interaction as touching, pawing, or biting the puzzle box. The proportion of time interacting with the puzzle box was used as a measure of persistence. We also recorded latency to approach, interact with, and touch the puzzle box as a second measure of boldness because the responses of coyotes to the novel object and puzzle box were likely to differ since we presented the puzzle box with food and across repeated trials; thus, providing measures of boldness in novel situations and foraging contexts. Finally, we used the number of solutions a coyote solved as a measure of repeated innovation and flexibility.

Conspecific aggression trials

We used a dummy coyote (Lone Howler Coyote Decoy, Flambeau Outdoors, Middlefield, OH, USA; Fig. S4) to test the response of coyotes to an unknown conspecific every three months starting in the pretrial phase (Fig. S2; Table S2). We placed the dummy near the edge of the enclosure and for one hour recorded how coyotes interacted with the dummy. We did not include any scent or auditory stimuli. A trained observer coded videos to record the latency for coyotes to approach within 5 m, 1 m, and interact with the dummy. We documented instances of aggressive or playful behavior directed towards the dummy (Table 1). We conducted the novel object, puzzle box, and conspecific tests every three months to provide longitudinal data without constantly exposing the coyotes to unfamiliar stressors.

Coder reliability

Live focal observations were initially conducted by a single observer before two other observers joined the study in March and August. Before conducting independent observations, new observers conducted at least 10 simultaneous observations with the original observer for training. Training continued until the total time spent in each activity exhibited >90% agreement. The same behavioral categories were used for video-recorded feeding observations, and all three trained observers coded feeding videos after successful training on live observations.

The same three observers coded novel object, puzzle box, and conspecific videos. We trained each coder by watching multiple videos together and at least one video independently to ensure consistency. We conducted a post hoc test of coder reliability with the original coder independently coding one puzzle box video from each of the other coders. The intraclass correlation was >0.95 between all video coders (95% Confidence interval: 0.76–1.00; McGraw & Wong, 1996). We conducted coder reliability on a puzzle box video because it contained all the same behavior definitions as novel object and conspecific videos.

Statistical analyses

We combined behaviors into five categories because of the low proportion of time spent in many activities: resting, active (locomoting and stereotyping), feeding (foraging, eating, investigating prey model, interacting with prey model), social (aggression, play, howling), and prey model (investigating prey model, interacting with prey model). For the feeding trials with the prey model, investigating and interacting with the prey model were included in feeding behavior, but also analyzed separately because of our interest in behaviors directed at the prey model. We did not include standing behavior in the analysis because it made up a small proportion of time and lacked ecological relevance. Initial plotting showed no trends in scent-marking behavior; therefore, we did not formally analyze these data. We used mixed-effects beta regression from the “glmmTMB” package (Brooks et al., 2017) in Program R version 4.1.0 (R Core Team, 2021) to test fixed effects of month and treatment and an interaction between month and treatment. We included random effects for individual coyote ID nested within pair. With this model specification, a significant interaction term indicates that treatment and control coyotes showed different behavioral changes through time, suggesting a treatment effect. The fixed effect of month accounts for seasonal shifts in behavior, and our control group allows us to separate seasonal trends from treatment effects. Our design also accounts for changes in behavior due to breeding since both control and treatment groups had equal numbers of breeding and nonbreeding pairs.

For novel object data, we used mixed-effects cox proportional hazards models from the ‘coxme’ package (Therneau, 2020) to test the effects of time (i.e., month of study), treatment, and their interaction on latency to approach 5 m, 1 m, and touch the novel object. We also included a random effect to control for pair and coyote ID. Each behavioral metric was tested in a separate model for all statistical tests. For the puzzle box data, we used mixed-effects beta regression to model the proportion of trial time that coyotes spent interacting with the puzzle box as a measure of persistence using the same model structure as above. We also used mixed-effects cox proportional hazards models to model the latency to approach 1 m and interact with the puzzle box as a measure of boldness. Because only a single coyote solved the puzzle box, we did not conduct any statistical analyses on solving data. For the conspecific test data, we also used mixed-effects cox proportional hazards models and the same model structure as above to model the latency to approach within 5 m and 1 m of the conspecific model. Due to limited interactions with the conspecific model, we did not conduct statistical analyses with interaction data.

We used principal component analysis (PCA) and permutational multivariate ANOVA (PERMANOVA) to evaluate multivariate trends in coyote behavior. We created a multivariate response matrix that included monthly mean proportions of time resting, feeding, and moving during both feeding and nonfeeding observations, monthly mean proportion of time spent within 1 m of the puzzle box, monthly mean proportion of time interacting with the puzzle box, and the latency to approach within 1 m of the puzzle box and 5 m of the novel object for each coyote. We plotted the first two components of the PCA for each month to visualize the grouping of hunting and nonhunting coyotes in multivariate space. We then used PERMANOVA to statistically evaluate separation between groups (Oksanen et al., 2018). We conducted a separate PERMANOVA for each month and included a fixed effect for foraging treatment. All statistical analyses were conducted in Program R 4.1.0 (R Core Team, 2021). We used the BORIS software to code all videos (Friard & Gamba, 2016).

Results

Feeding and nonfeeding focal sampling

Few coyotes successfully hunted the prey model (Table S3). Ten individual coyotes interacted with the prey model, moving or stationary, at least once. Only six coyotes regularly fed from the prey model, two hunting coyotes and four nonhunting coyotes. The two hunting coyotes were from different pairs, one male and one female, and began regularly feeding from the moving prey model in February and April. The four nonhunting coyotes were from three different pairs, two females and two males, and began regularly feeding from the stationary model in May, June, July, and September (Fig. S2). Despite this lack of hunting, we analyzed the data using the a priori treatment assignments. This approach should lead to conservative estimates of change due to the treatment, as hunting coyotes that did not interact with the prey model should weaken treatment effects.

During nonfeeding observations, both hunting and nonhunting coyotes increased the proportion of time spent resting throughout the study (z = 2.284, p = 0.022). This increase was greater for hunting coyotes (z = 2.10, p = 0.036; Fig. 1). The increase in resting behavior was paired with a decrease in time spent active for both groups (temporal term: z = −3.459, p = 0.001; temporal-treatment interaction term: z = −2.302, p = 0.021). Stereotyping behavior decreased in hunting coyotes, but not to a significant degree (z = −0.553, p = 0.580). Contrary to our predictions, there were no significant trends in time spent foraging by either hunting or nonhunting coyotes during nonfeeding times (Table S4).

Figure 1 Nonfeeding behavior.

The proportion of time spent resting by hunting coyotes (grey triangles) and nonhunting coyotes (black circles) during nonfeeding focal observations throughout the study period from November 2019 (Pre) to November 2020. Error bars represent standard error.

During feeding observations, both hunting and nonhunting coyotes increased the proportion of time they spent feeding as the study progressed (z = 3.025, p = 0.002), which was paired with a decrease in time spent active outside of feeding (z = −8.611, p < 0.001; Fig. 2). We did not observe any changes in other behaviors during feeding observations and there was no evidence of treatment effects on behavioral budgets during feeding observations (Table S5).

Figure 2 Feeding behavior.

The proportion of time that hunting (grey triangles) and nonhunting (black circles) coyotes spent feeding during feeding trials from November 2019 (Pre) through November 2020. Error bars represent standard error.

During prey model feeding observations, hunting coyotes decreased the proportion of time spent feeding throughout the study (z = −3.945, p ≤ 0.001) while nonhunting coyotes decreased the proportion of time spent active (z = −7.762, p ≤ 0.001). Both hunting and nonhunting coyotes increased the proportion of time interacting with the prey model (z = 2.394, p = 0.017), and this increase was larger for hunting coyotes (z = 2.522, p = 0.012; Fig. 3; Table S6). The two coyotes that successfully hunted the prey model displayed progressive innovation in solving the foraging task. Initially, they followed the prey model, consuming small bits of food that fell off when the model hit bumps. They then progressed to grabbing mouthfuls of food off the moving model, before ultimately developing strategies of biting or pawing the model to knock it over. While they were able to access some food without innovating, modifying behaviors increased the amount of food available.

Figure 3 Prey model behavior.

The proportion of time that hunting (grey triangles) and nonhunting (black circles) coyotes spent interacting with the prey model during feedings from January through November 2020. Error bars represent standard error.

Novel object trials

For the novel object, puzzle box, and conspecific tests, results were similar at the different distances observed (5 m, 1 m, touch/interact). We present representative results here, and model results for each distance are in the supplementary material. The initial presentation of the puzzle box, without food, was used as a novel object trial. All subsequent novel objects were different objects.

In general, all coyotes showed an unwillingness to approach novel objects. Only five coyotes approached within 1 m of a novel object, and only one regularly approached novel objects (Table S3). Of 78 recorded trials, coyotes approached the novel object within 5 m 31 times, 1 m eight times, and touched only twice. There were no significant effects of time, treatment, or their interaction on the latency to approach 5 m, 1 m, or touch the novel object (Table S7).

Multi-access puzzle box tests

Seven coyotes interacted with the puzzle box at least once, but four of these did not occur until August or November. Only a single coyote successfully solved any of the puzzle box solutions (Table S3). This was a hunting coyote that solved two doors in August and one door in November. Because of this lack of solving, we were unable to use the puzzle box as an indicator of innovation in coyotes. Both hunting and nonhunting coyotes increased the proportion of time they spent within 1 m of the puzzle box throughout the study period (z = 2.292, p = 0.022), and this increase was greater for hunting coyotes (z = 4.271, p < 0.001). Neither group showed any trend in the proportion of time interacting with the puzzle box (Fig. 4A; Table S8). Both hunting and nonhunting coyotes decreased their latency to approach within 1 m of the puzzle box throughout the study (z = 2.78, p = 0.006). Only hunting coyotes decreased their latency to interact with the puzzle box (z = 2.59, p = 0.010; Fig. 4B; Table S9).

Figure 4 Puzzle box behavior.

(A) The proportion of time that hunting coyotes (grey triangles) and nonhunting coyotes (black circles) spent within 1 m of the puzzle box and (B) the latency of coyotes to interact with the puzzle box during puzzle box trials throughout the study period from November 2019 (Pre) through November 2020. Error bars represent standard error.

Conspecific aggression trials

Coyotes showed minimal interest in the coyote dummy used in the conspecific test. Fourteen coyotes approached within 5 m of the conspecific dummy, but only four did so multiple times (Table S3). During the pretrial phase, one coyote pulled the dummy down by the tail, but no other direct interactions occurred throughout the study. Both hunting and nonhunting coyotes slightly decreased their latency to approach within 5 m of the dummy across time (z = 2.00, p = 0.045), but there were no trends in latency to approach within 1 m (z = −1.19, p = 0.23; Table S10).

Combined results

Only a single coyote displayed measurable behaviors in all tests (Table S3). This was a hunting coyote who began regularly capturing the prey model in April, approached within 1 m of four novel objects, approached within 5 m of the conspecific model on four occasions, and solved two doors on the puzzle box in August and one door in November. This was also the only coyote involved in the study that had two wild-born parents.

Multivariate analysis revealed that hunting and nonhunting coyotes occupied similar behavioral spaces until the end of the study. The first two components of the PCA explained 69% of the variation in behavioral space (Table 2). Visually, the PCA plots show overlap between the two treatment groups in the pretrial, February, May, and August data. However, the November plot shows some separation between groups. PERMANOVA confirmed that there were significant differences in behavioral space for hunting and nonhunting coyotes in November (F = 4.024, p = 0.018; Fig. 5; Table S11). Based on PCA loadings, this difference was largely due to hunting coyotes spending more time resting during focal observations.

Table 2 Variable loadings and proportion of variance explained by the first two components of the principal component analysis.

The principal component analysis (PCA) loading for each behavioral metric included in the PCA for the first two components. Behavioral metrics followed by (NF) are from nonfeeding observations and metrics followed by (F) are from feeding observations. The final row provides the proportion of variance explained by each component.

Behavioral metric	PC1 loading	PC2 loading	
Prop time rest (NF)	−0.5466	0.2456	
Prop time feed (NF)	0.4099	−0.0494	
Prop time active (NF)	0.5205	−0.0989	
Prop time rest (F)	0.0176	−0.0461	
Prop time feed (F)	−0.3777	0.0559	
Prop time active (F)	0.2746	0.1379	
Puzzle box w/in 1 m	−0.1428	−0.5071	
Puzzle box interact	−0.0206	−0.0456	
Novel object latency 5 m	0.0493	−0.2328	
Puzzle box latency 1 m	0.1438	0.7769	
Proportion of Variance Explained	0.491	0.687	

Figure 5 PCA results.

Plots of the first and second principal component analysis (PCA) axes for hunting (grey triangles) and nonhunting (black circles) coyotes for the pre-treatment data (A), in February (B), August (C), and November (D).

Discussion

We used a controlled experiment to determine how changes in foraging behavior affect individual behavior in captive coyotes. We documented changes in behavioral budgets, boldness, and persistence of coyotes given the opportunity to pursue prey compared to those that remained as scavengers. Consistent with our predictions, hunting coyotes increased boldness and persistence during the puzzle box tests and hunting trials, respectively, and increased time spent resting, which was partially due to decreases in time spent locomoting and stereotyping. Contrary to our hypotheses, we did not observe increases in time spent foraging, innovation, or response to a conspecific. These results indicate that shifts in foraging behavior affect some aspects of predator behavior but not others. Our results are similar to findings from wild black bears (Ursus americanus), which decreased activity and changed temporal activity patterns with changes to foraging behaviors (Beckmann & Berger, 2003). This may have broader consequences, such as shortening hibernation season and altering reproductive patterns of bears (Johnson et al., 2018; Gould et al., 2021). The wild bears shifted from hunting to increased scavenging, while we implemented the opposite shift with coyotes, yet the same pattern of foraging behavior altering other behavioral traits occurred. Thus, this trend could exist across predators and requires further investigation in wild and captive settings.

Changes to coyote behavioral budgets could reflect the different mechanisms that facilitate successful foraging by predators who hunt versus scavenge. Hunting predators often pursue prey during times that optimize success and rest otherwise (Kohl et al., 2019), while scavenging predators must cover large areas in search of food and are therefore more active across time (Kane & Kendall, 2017). In our system, scatter feeding likely left small pieces of food more dispersed throughout the enclosure than our hunting treatment. Scavenging coyotes may have been more likely to search for remaining pieces throughout the day and, thus, remained more active. However, we did not observe changes in the proportion of time foraging, which would have provided additional support for this hypothesis. While delivering food unpredictably in space and time increases food searching behavior in captive red foxes (Vulpes vulpes; Kistler et al., 2009), it does not have the same effect on captive coyotes. Instead, captive coyotes change only a few behaviors, such as vocalization and marking, along with the time of day certain behaviors occurred (Gilbert-Norton, Leaver & Shivik, 2009). Thus, adding temporal unpredictability in addition to hunting behavior could alter predator behavior synergistically. Scavenged resources are often more temporally predictable than live prey, potentially impacting behavioral budgets in wild predators (Mourier, Claudet & Planes, 2020). Predator activity patterns that may change to better track novel resource use can have cascading effects on other wildlife by altering the non-consumptive effects of predators (Moll et al., 2017). Prey species and mesopredators often seek refuge in both space and time to minimize predation risk (Kohl et al., 2019; Smith et al., 2020). Altering predator activity budgets could shift when and how frequently prey are released from predation risk.

We also observed seasonal trends in the behavioral data, including changes in resting during nonfeeding observations (Fig. 1) and time spent feeding during feeding observations (Fig. 2). We expected seasonal trends in behavior (Ellington, Muntz & Gehrt, 2020), and some of these trends were likely related to the breeding season. Our treatment-control design allowed us to account for these seasonal trends in analysis. We did not observe any clear differences in the response to the puzzle box or novel objects by coyotes raising pups either during observations or in the data. Because we designed the project to evaluate foraging behavior and due to small sample sizes, we were unable to explicitly examine changes related to breeding season or between breeding and nonbreeding coyotes.

Persistence is likely a key trait for predator success both while hunting and when exploiting scavenged foods. Successfully capturing prey requires identifying, pursuing, and attacking target animals (MacNulty, Mech & Smith, 2007), and cursorial predators may cover multiple kilometers through this process (Hubel et al., 2016). Persistence is required because successful hunts are rare (Cresswell & Quinn, 2010; Hubel et al., 2016). Persistence is also required for scavenging predators due to the risks presented by other predators and humans (Prugh & Sivy, 2020) or having to access novel food resources (Young, Touzot & Brummer, 2019). In our system, hunting coyotes had to learn to pursue and capture a novel prey item. Completing this task required multiple exposures and persistence, and most coyotes did not begin regularly interacting with the prey model until the fifth month of trials (Fig. 2). Hunting coyotes showed a greater increase in time spent interacting with the prey model, indicating higher persistence in the foraging task.

Persistence is also a key trait for scavenging predators learning to access novel food resources, particularly in urban environments where food may be secured in trash bins and containers, behind fences, or inside structures (Beckmann & Berger, 2003; Newsome et al., 2015a; Newsome et al., 2015b; Breck et al., 2019). However, we did not observe changes in persistence during interactions with the puzzle box. Most coyotes approached more quickly and spent more time near the puzzle box as the study progressed but did not spend more time interacting with the puzzle box. This contradiction between persistence during the hunting task and the puzzle box trials could be related to the more visibly accessible food during hunting trials or because coyotes were exposed to the prey model every week instead of every three months thereby increasing habituation. Hunting and scavenging predators are both likely to experience new foraging tasks in the form of novel prey or non-prey foods due to global change. These changes could introduce additional selective pressures on predators for persistence due to links between persistence and problem solving (Chow, Lea & Leaver, 2016; Daniels et al., 2019; Young, Touzot & Brummer, 2019).

We documented increased boldness with respect to the puzzle box in both groups, but not toward the novel objects. This is likely because coyotes associated the puzzle box with a food reward and habituated to its repeated presence compared to the novel objects, and were, therefore, more willing to approach as the study progressed. Previous research has identified boldness as a key trait for urban predators (Breck et al., 2019; Brooks, Kays & Hare, 2020). Bold predators are more likely to approach and interact with novel objects and less likely to be disturbed by human presence, thus increasing the likelihood of discovering new foraging opportunities (Young et al., 2020). However, bold predators may also experience reduced survival because they are more likely to incite human-wildlife conflict (Greenberg & Holekamp, 2017). Therefore, boldness may not only correlate to accessing resources but also human perceptions of wildlife (Young et al., 2020).

The ecological and conservation consequences of persistent and bold predators are complex. Bold predators are more prone to human-wildlife conflict and more likely to be lethally removed due to conflict situations (Bombieri et al., 2021; Schell et al., 2021), and increased predator boldness could hamper predator conservation efforts in wildland-urban interfaces (Nyhus & Tilson, 2004; Brooks, Kays & Hare, 2020). Despite this, the prevalence of increased boldness in predators expanding their range into urban environments indicates boldness is a valuable trait in novel environments (Barrett, Stanton & Benson-Amram, 2019). Boldness also impacts individual foraging decisions. Bold individuals are more likely to forage in dangerous habitats, which may alter available food resources (Toscano et al., 2016). For mesopredators, increased boldness could also increase predation risk (Geffroy et al., 2015), disrupting expected niche partitioning behaviors. Knowledge on the effects of traits like boldness or persistence on hunting success is lacking and should be addressed in the future to better understand how behavioral traits impact predator–prey interactions.

The captive nature of the study subjects allowed necessary control over experimental conditions, yet inherently limits the scope of inference and the transferability to wild predators. Because the subjects were housed as mated pairs in enclosures, they were insulated from the effects of direct intra- and interspecific interactions, dynamic environments, and other aspects of natural systems. They also likely responded to each other during our trials, although we never observed more than one individual within a pair participate in any trials. While we do not think the individuals within a pair prevented the other from participating, and wild coyotes typically maintain pair bonds, further tests of individuals could be warranted to compare with our results. Regardless, supporting the concept of linkages between foraging mode and behavior with captive animals is a valuable first step. Additionally, our approach shifted scavenging predators to become hunting predators, the opposite shift we hypothesize to become more common with global change (Parsons, Newsome & Young, 2022). However, the broad relationship between foraging mode and other behavioral traits we observed likely exists across predators, whether they are captive or wild. This is supported by evidence of behavioral change in urban black bears with access to human refuse (Beckmann & Berger, 2003) and multiple species with access to landfills (Hidalgo-Mihart et al., 2004; Cozzi et al., 2016). Our results indicate that activity budgets, boldness, and persistence may change with altered resource use, but specific changes will be dependent on the context of predator species and resource changes. We encourage future efforts with wild predators in areas with changing resource availability to evaluate behavioral responses.

Many predators successfully adapt to shifts in resource availability (Bateman & Fleming, 2012; Gámez & Harris, 2021). However, certain individuals in a population likely succeed in this adaptation, while others do not (Schell et al., 2021). In this study, only two of eight coyotes in the hunting treatment fully adapted to the foraging task and regularly fed from the prey model, while six remained wary of the prey model and only occasionally knocked it over. Our approach specifically addressed changes in behavior of individual coyotes over a short time. However, the highly variable participation of individual coyotes highlights that each had different baseline behavioral norms. The individuals who were bolder from the beginning were most likely to approach the prey model sooner and learn the new foraging task. Individuals who failed to adjust quickly would likely struggle in response to shifting resources, highlighting that shifts in resource availability can be a strong selection pressure for inherently bold, innovative, persistent, or flexible individuals. The understanding of the heritability of these behavioral traits is still developing, but research has identified genetic correlations with boldness in multiple vertebrate species (Bubac et al., 2021; Stratton, Nolte & Payseur, 2021) and differential heritability among different behaviors in donkeys (Equus asinus; Navas González et al., 2019). Therefore, environmental changes that select for bold, innovative, or persistent individuals paired with the heritability of these traits could result in shifts in behavior at the population level over time. These selection patterns, in addition to resource spatial heterogeneity, could lead to segregation of predators with different traits and the possible development of new ecotypes (Chilvers & Corkeron, 2001).

There were limited interactions between coyotes and the puzzle box and conspecific model. Like previous research, coyotes struggled to solve the puzzle box task (Stanton et al., 2021). Only one coyote solved the puzzle box, indicating that it was possible but that most of the coyotes were unwilling to interact with the puzzle box enough to find solutions. Very few coyotes made significant attempts to access the food inside the puzzle box and were content to eat the small portion scattered around the box even though the quantity was insufficient to reach satiation. Therefore, we were unable to draw conclusions related to innovation. Similarly, few coyotes interacted with the coyote dummy for us to measure conspecific interactions. We observed many coyotes approach the dummy at the first encounter, but few approached within 5 m of or interacted with the dummy. It is likely that as the coyote approached the dummy, they recognized it was inanimate and not a territorial intruder. Adding a scent or auditory cue may have increased interactions, but we elected to keep methods consistent throughout this study. For animal welfare purposes, we could not use a live conspecific coyote; properly documenting interactions with unfamiliar conspecifics would best be observed in the wild.

Conclusions

Changes to resource availability due to global change will continue to alter predator foraging behavior. It is well understood that changes in foraging behavior can impact space use (Hidalgo-Mihart et al., 2004), migration (Cozzi et al., 2016), and social structure (Chilvers & Corkeron, 2001). Individual behavioral changes are less well understood, likely because it is difficult to observe in the wild due to limited ability to repeatedly test or alter available resources for individuals. Working with captive animals allowed us to observe individuals succeed and fail at adapting to a new foraging task and showed how some behavioral traits change in response to altered foraging behavior. Altered activity budgets, persistence, and boldness will affect how predators interact with other wildlife and humans, leading to ecological consequences (Parsons, Newsome & Young, 2022) and increased human-wildlife conflict (Barrett, Stanton & Benson-Amram, 2019). Future research with wild predators should evaluate the relationship between predator behavioral traits and the proportion of natural food in diets to further elucidate links between foraging and behavior and improve our understanding of ecological consequences.

Supplemental Information

Supplemental Information 1 Supplemental text, figures, and tables

Click here for additional data file.

We thank S Brummer, M Davis, N Floyd, S Keller, A Merical, and J Shultz for assistance with animal care and research logistics. C Dever, T Khvtisiashvili, and K Pelaez helped in coding observation videos.

Additional Information and Declarations

Competing Interests

Author Contributions

Animal Ethics

Data Availability

The authors declare there are no competing interests.

Mitchell A. Parsons conceived and designed the experiments, performed the experiments, analyzed the data, prepared figures and/or tables, authored or reviewed drafts of the paper, and approved the final draft.

Andrew Garcia performed the experiments, analyzed the data, authored or reviewed drafts of the paper, and approved the final draft.

Julie K. Young conceived and designed the experiments, performed the experiments, prepared figures and/or tables, authored or reviewed drafts of the paper, and approved the final draft.

The following information was supplied relating to ethical approvals (i.e., approving body and any reference numbers):

All research methods and procedures were approved by USDA-National Wildlife Research Center’s Institutional Animal Care and Use Committee (QA-3151).

The following information was supplied regarding data availability:

The raw data are available at GitHub: https://github.com/pars2997/scavenging-hunting-coyotes.

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
