# Peer review of "Scavenging vs hunting affects behavioral traits of an opportunistic carnivore"

_PeerJ, doi:10.7717/peerj.13366_

## Round 0.1 · original submission · Major Revisions

I received three detailed and thoughtful reviews of your work with each reviewer pointing to slightly different issues with your work. Therefore, I am classifying this as a major revision, even though two of the three reviewers independently identified the required changes as minor. I think that the changes are doable though and find it likely that a thorough revision will result in a publishable paper. I agree with the reviewers’ comments and believe that the clarity they are asking for will increase the value of your work. Importantly, all three reviewers and I already find the work valuable and recognize the challenges inherent in this kind of work. I commend you on asking an important question about the potential effects of human-imposed changes in foraging opportunities for coyotes. I am also glad to see that you recognize the limitations as a function of testing these associations in captive coyotes.

I have just a few comments of my own:

In the natural model you are interested in, coyotes may be natural hunters but may turn to increased scavenging because of changes in food sources, whereas, in your captive model, the change is moving in the opposite direction - from scavenging entirely to hunting being a novel activity. You note this in passing around line 308 and again a bit later in the discussion, but I think this confound should be addressed directly. The fact that hunting is novel for your captive coyotes (as I understand it) is a threat to the ecological validity of your design and needs to be addressed upfront. Coyotes would not typically be exclusive scavengers or hunters, would they? The reviewers have also asked for more detail on the history of your coyotes, which is relevant here. There are also limits on options to hunt in captivity that do not exist in wild coyotes. Your captive coyotes may be particularly motivated to hunt the new prey because of the novelty or otherwise lower levels of stimulation compared to wild coyotes. Similarly, hunting is not a necessity because they are presumably well fed in captivity. So, I think more needs to be said to justify the use of captive coyotes to investigate this research topic. You have increased control, but to some degree, this control may be more of a cost than a benefit to your study since it does not seem natural to create coyotes that are exclusively hunters or scavengers.

Can you say more about how your prey model created new opportunities for problem solving. Unless the prey model was able to escape, I am not sure how much problem solving the coyotes had to engage in to “hunt” it. Persistence sometimes involves continuing to work on problems that one cannot solve. If this task is always solvable, persistence should be selected for, but this again does not mimic foraging or hunting strategies in the wild.
How was persistence factored into your analytic model since it could not be compared between groups? Was it a mediator or moderator variable? I think the variables and their role in your model need to be clearer before you get to your Methods.

Why did you not counterbalance the order of presentation of novel objects? It is difficult to be certain that changes over time were due to increased boldness or differing levels of threat perceived from different stimuli in this case.

In the initial phase of the puzzle box trials, why did you remove the food and open all the of the doors? Would you not want the coyotes to strongly associate the box with the delivery of food? Would you not want to get them comfortable retrieving food from the box by giving them multiple opportunities to do so? Did you film continuously for the entire two-hour trials? Otherwise, how did you know which solution was used because once the food was removed, coyotes might have used an additional solution to open a door even if the food was already removed. Why was a solution locked after only a single use? That makes it less likely to observe persistence or perseveration. Usually the measure of innovation is taken as using a novel solution after one solution is used consistently and then becomes inoperable. Why did you open all solutions again in the subsequent test period? More of these decisions require justification.

Innovation can refer to the application of a novel solution to an old problem as well – not just to novel problems (line 66).
Given the lack of response to the dummy coyote, you might have considered replacing with a different stimulus, such as the scent or a playback of unfamiliar coyotes.

In the paragraph beginning on line 392, you discuss how shifts in resource availability can be a selection pressure but these effects are likely to happen at the population, rather than at the individual level. I think more needs to be said here to justify the focus on changes within individuals within their lifetime compared to the selection pressures one expects to exert an influence on populations over time.

Line 75, insert the authors’ name and publication date here instead of [23].

You might find the following papers valuable too:

Rodriguez, J. T., Lesmeister, D. B., & Levi, T. (2021). Mesocarnivore landscape use along a gradient of urban, rural, and forest cover. PeerJ, 9, e11083.

Sol, D. (2009). Revisiting the cognitive buffer hypothesis for the evolution of large brains. Biology letters, 5(1), 130-133.

Reviewer 1 ·

Basic reporting

a. The language of this paper is clear and concise. The Introduction provides a broad overview of why carnivores may shift foraging strategies from hunting to scavenging, and how behavioral changes may accompany this shift. I think the authors should include more emphasis on how and why anthropogenic resources/urbanization leads to scavenging specifically while not also increasing small prey items to be hunted such as mice and birds.
b. I am not convinced that an increase in scavenging would necessarily lead to a decrease in problem solving ability. If scavenging involves, for example, deliberate extractive foraging, this may provide selection pressure to enhance problem solving ability and innovation in scavenging individuals/species (Liker & Bokony, 2009; Auersperg et al., 2011; Griffin & Guez, 2014; Goodman, Hayward & Hunt, 2018; Barrett, Stanton & Benson-Amram, 2019; Daniels et al., 2019).
c. The figures are simple, clear, and easily interpreted by the reader.
d. The raw data and analysis code have been supplied on github. It appears to be complete, and has the appropriate accompanying metadata.

Auersperg AMI, von Bayern AMP, Gajdon GK, Huber L, Kacelnik A. 2011. Flexibility in Problem Solving and Tool Use of Kea and New Caledonian Crows in a Multi Access Box Paradigm. PLoS ONE 6:e20231. DOI: 10.1371/journal.pone.0020231.
Barrett LP, Stanton LA, Benson-Amram S. 2019. The cognition of ‘nuisance’ species. Animal Behaviour 147:167–177. DOI: 10.1016/j.anbehav.2018.05.005.
Daniels SE, Fanelli RE, Gilbert A, Benson-Amram S. 2019. Behavioral flexibility of a generalist carnivore. Animal Cognition 22:387–396. DOI: 10.1007/s10071-019-01252-7.
Goodman M, Hayward T, Hunt GR. 2018. Habitual tool use innovated by free-living New Zealand kea. Scientific Reports 8:13935. DOI: 10.1038/s41598-018-32363-9.
Griffin AS, Guez D. 2014. Innovation and problem solving: A review of common mechanisms. Behavioural Processes 109:121–134. DOI: 10.1016/j.beproc.2014.08.027.
Liker A, Bokony V. 2009. Larger groups are more successful in innovative problem solving in house sparrows. Proceedings of the National Academy of Sciences 106:7893–7898. DOI: 10.1073/pnas.0900042106.

Experimental design

The research questions are clearly defined and meaningful. There are a few areas where more detail should be provided in order to replicate this study:
a. Please include more information on the coyotes participating in this study. Were they all captive reared? How many generations have been in captivity? What generation is this? Do the coyotes ever have the opportunity to hunt within their enclosures (i.e., prey accidentally wanders in)? This information would help readers interpret how much of the results of this study could be extrapolated to wild coyote or carnivore populations.
b. I suggest the authors report how many total observations/observation time totals to summarize for the reader. It would also be helpful to include a table or timeline outlining when and in what order each trial was conducted, as this is a lot for the reader to synthesize on their own.
c. Were any odor cues provided when the dummy coyote was presented? Please include whether odor was used as an additional stimulus in these trials.
d. What defined a ‘capture’ of the car prey item?

Validity of the findings

a. My major concern is the validity of time spent within 1m of the puzzle box as a measure of persistence. Most studies I am aware of that measure persistence in problem solving use a measure of time spent interacting with the puzzle box, or a rate of attempts at problem solving (Griffin & Guez, 2014; Chow, Lea & Leaver, 2016; Brubaker et al., 2017; Jacobson et al., 2021). I think the more likely explanation for the change in the persistence measure in this study is habituation to the puzzle box, which is supported by the fact that both groups increased the amount of time spent within 1m of the puzzle box over time and therefore number of presentations of the puzzle box. This is what I would predict would happen as the coyotes habituate to the novel object (puzzle box), and is what the authors discuss in regards to boldness in lines 354-356. I think this adds to your results of boldness differences between the two groups, since the hunting coyotes had a greater increase in this measure, but am skeptical that any valid conclusions can be made regarding persistence from this measure alone.

b. Lines 394-397 should be included in the Results section, as this is quite a small sample that successfully adapted to the foraging device, and limits conclusions that can be inferred from these data. Results should be updated to include the total numbers of individuals that successfully participated in all trials.

c. Could the change in activity budgets (lines 246 – 248) be attributed to natural seasonal changes in activity rather than the foraging treatment?

d. I largely agree with the rest of the results and interpretation presented by the authors.

Brubaker L, Dasgupta S, Bhattacharjee D, Bhadra A, Udell MAR. 2017. Differences in problem-solving between canid populations: Do domestication and lifetime experience affect persistence? Animal Cognition 20:717–723. DOI: 10.1007/s10071-017-1093-7.
Chow PKY, Lea SEG, Leaver LA. 2016. How practice makes perfect: the role of persistence, flexibility and learning in problem-solving efficiency. Animal Behaviour 112:273–283. DOI: 10.1016/j.anbehav.2015.11.014.
Griffin AS, Guez D. 2014. Innovation and problem solving: A review of common mechanisms. Behavioural Processes 109:121–134. DOI: 10.1016/j.beproc.2014.08.027.
Jacobson SL, Puitiza A, Snyder RJ, Sheppard A, Plotnik JM. 2021. Persistence is key: investigating innovative problem solving by Asian elephants using a novel multi-access box. Animal Cognition. DOI: 10.1007/s10071-021-01576-3.

Additional comments

Please expand your ethics statement to include more information on the care and use of the study participants.

Reviewer 2 ·

Basic reporting

The manuscript is clearly written and unambiguous, and uses clear, technically correct text while conforming to professional standards of courtesy and expression.

The introduction and background are sufficient to determine how the work fits into the broader field of knowledge, and relevant literature is appropriately and extensively referenced.

The structure of the article conforms to an acceptable format of ‘standard sections’, and the figures presented are relevant to the content of the article, of sufficient resolution, and appropriately described and labeled. The raw data is posted to GitHUB.

The manuscript is self-contained with relevant results to hypotheses and thus is an appropriate ‘unit of publication’. The manuscript also effectively identifies directions for additional research and follow-up.

I do have some comments/questions related to the Methods and Discussion, which are detailed below.

Experimental design

The manuscript contains original primary research that falls within the Aims and Scope of the journal, and is relevant to both Biological Sciences and Environmental Sciences. The research question is well defined, relevant to the biological and environmental sciences, and meaningful. The submission identified the knowledge gap being investigated and makes clear statements on how the study contributes to filling that gap.

The experiments and investigation were rigorously conducted in conformity with prevailing ethical standards for animal care, and the methods have been described with sufficient information to be reproducible by other investigators.

Validity of the findings

The manuscript does not report on research that is redundant or derivative of existing work.

The authors have provided all underlying data in an acceptable discipline-specific repository (https://github.com/pars2997/scavenging-hunting-coyotes).

Conclusions are well stated, linked to original research question & limited to supporting results. I do have some requests for some additional details and context.

Additional comments

The manuscript is comprehensive, well-written, appropriately referenced, and uses appropriate methods for data collection and analyses. I have mostly minor comments.

Introduction

The Introduction does an excellent job framing the research, and hypotheses and predictions are clearly stated

- lines 68-70 - while I agree with this statement and think it's accurate, can any supporting references be added?

- line 75 - different citation format, what reference supports this statement?

- line 76 - I would change "carnivores" here to "predators", since bottlenose dolphins are used as an example (they are carnivorous but not "carnivores" per se)

Materials & Methods

lines 122-127 - The study used 16 coyotes (8 pairs) out of approximately 90 adult coyotes that are maintained as male-female pairs in outdoor enclosures ranging from 0.1 – 1.0 ha. How close are the enclosures to one another, and could adjacent pairs influence the behaviour of the coyotes used in the study? If yes, how was this addressed?

lines 125-126 - "Coyotes in this facility display similar behavioral budgets to wild coyotes (Shivik et al., 2009)"
- did the behavioral budgets in Shivik et al. (2009) use similar methods as the ones developed in this study? (timing, bout length, etc.)

lines 128-130 - The study overlapped the coyote mating and pup rearing seasons from February to June, and half the coyote pairs (two treatment pairs, two control pairs) bred during the study period. I am curious as to whether breeding behavior could have influenced other coyote behaviors, and whether adjacent animals are close enough to one another to react in any significant way. (see additional comments on this below)

- lines 148-149 - "Initially, we left the prey model stationary and scattered additional food around it for coyotes to become comfortable with the prey model and associate it with food."
- how long was this initial period?

lines 153-155 - "If coyotes did not successfully hunt on a given day, we removed the prey model and returned at least two hours later and fed the pair using standard procedure for the facility."
- how often did this happen, and was it considered in the analyses?

162-165 - "We modified the ethogram from Shivik et al. (2009; Table 1) and conducted continuous observations for 15 minutes for each individual coyote. Each month, we observed each coyote during four non-feeding times, four feedings without the prey model, and four feedings with the prey model."
- did the original study use the same methods?
- is their support in the literature for the use of 15 minute observation bouts? Any data on sensitivity of ethogram results to longer or shorter observation times?

line 202-204 - the coyote dummy was visual only, correct? No coyote urine, etc. was used? (see additional comments on this below)

Statistical Analyses
211-213 - For behavioral budget data, we combined behaviors into four categories: resting, active (locomoting and stereotyping), feeding (foraging and eating), and social (aggression, play, howling).
- Table 1 says "Moving" and not "active" for the broad category
- how was proportion of time spend standing or scent marking considered in analyses, since these behaviors are included in the ethogram but not listed here?
- the sentence should also include the interactions with the prey model (as noted in Table 1)

Results

lines 254-256 - "During feeding observations, both hunting and nonhunting coyotes increased the proportion of time they spent feeding (z = 3.025, p = 0.002) which was paired with a decrease in time spent active but not feeding (z = -8.611, p < 0.001; Figure 2)."
- we would expect an increase in time spent feeding when they are being fed, what about other behavior categories besides "active"?

Discussion

lines 379-380 - "The subjects were insulated from the effects of intra- and interspecific interactions, dynamic environments, and other aspects of natural systems."
- some additional context is needed here on how subjects were insulated from other coyotes
- for example, were study pairs kept separate from other animals at the facility, and if so, how?

line 382 - change "approached" to "approach"

lines 394-396 - "In this study, only two of eight coyotes in the hunting treatment fully adapted to the foraging task and regularly fed from the prey model."
- one pair, or two individuals from different pairs?

lines 408-414 - what about adding scent (i.e., coyote urine) to the dummy?

Conclusions

- line 425 - is Parsons et al. 2021 or 2022? 2021 is used here vs 2022 elsewhere

- line 426 - remove period after "conflict"

Some of the biggest differences between hunting and nonhunting coyotes occurred in May and June (e.g., proportion of time spent resting in Figure 1, proportion of time spent interacting with the prey model in Figure 3). What effects, if any, might the coyote's breeding status (e.g., rearing pups) have had on these findings?

Reviewer 3 ·

Basic reporting

The manuscript is well written, formatted clearly, and easy to follow. The authors have provided significant background for the context of the study and cited the relevant carnivore literature well. I have a few detailed comments for parts of the introduction that could be clarified or further expanded. In particular, I think there should be more acknowledgment of the variety of ways that some traits have been defined in the previous literature to distinguish exactly how they are being defined here. This is relevant for boldness which has often been separated from responses to novelty (see Carter et al., 2012). Since the terminology has varied widely in the literature, I think it is ok to use the term boldness in this study but it must be defined clearly and how it may differ from other studies using the same term should be outlined.

Experimental design

Some methodology needs clarification or further definition, but otherwise methodology and analysis appears appropriate.
It could be helpful to have a table with the traits studied, how you are defining each trait and the measures used specifically so the reader can keep track of the various measures for each trait. An images of the coyote dummy would also be helpful to include in supplementary materials. An important missing part of the methodology is more detail about who was conducting the behavioral observations or coding the video and measures of coding reliability for these observers. For the focal sampling, were the behaviors from the ethogram coded live? By one or multiple observers? How was this observer determined to be reliable? And for the video coding, there should be a portion of the video doubly coded for a measuring reliability of the trained observer to record latencies and durations.

Validity of the findings

The results and conclusions of this study are valid. The only piece I think may need to be supported more thoroughly or potentially reframed is whether the coyotes changed their foraging behavior or were provided with the opportunity to hunt but perhaps did not truly change a lot of their behavior. I know you present data on their proportion of interaction with the prey model over the course of the study but I still think it's not really clear how many of the coyotes were hunting. Potentially adding data about the number of times coyotes had to have food scattered after the prey model was provided could help support whether the behavior was changing. If you agree that there was not a lot of change in behavior, as it appears based on lines 394-6, then I think this needs to be discussed. For example in lines 303-304, you could address that perhaps the coyote hunting behavior didn't change enough to affect other behavioral changes, explaining some of the results. I still think even if the hunting wasn't adapted by all coyotes these results are still valuable, especially since you see the interaction effects for some measures. Perhaps language about the change that is happening should also be altered to say that the coyotes were presented with the opportunity to hunt, such as in line 298. I think with this issue discussed further, the conclusions are appropriate for the results.

Additional comments

Overall I commend the authors on this longitudinal multifaceted investigation of coyote behavioral traits and changes based on coyotes being provided with the opportunity to engage in hunting behavior to obtain food. I think the study has a strong between and within subjects design with appropriate controls for the non-hunting group.
Line by line comments
51-52- It would be helpful here to expand on the non-urban effects or provide a couple examples on how humans effect in non-urban ecosystems
56- predators “use of” alternative resources
60- I think you need to briefly introduce animal personality here with a definition before jumping into the traits
64-66- I would argue that persistence doesn’t necessarily need to be with new stimuli and could be applied to interactions with any foraging
70- I think you should clarify the trait changes for these particular traits here. I’m assuming you mean population-level changes with bolder, more persistent and more innovative individuals.
75- citation should be changed for Peer J format instead of number
123-4- It may be helpful to compare this housing to the species-typical social grouping of coyotes for those readers that may not be familiar with their behavioral ecology
128- overlapped “with” the
129-30 – this would be a good point to address in the Discussion, I would imagine that having pups around could have a significant affect on the behavioral responses to novelty and the adults’ interest in the puzzle box. If differences in the behavior of these pairs or lack of differences were observed please briefly describe in the Discussion
148-148- About how long did this phase last? Was it the same for every coyote or dependent on their behavior?
167-9- It may be more clear to move this information to before the detail about the persistence measure after introducing that you observe 4 non-feeding times, as on my first read I thought you were describing a set of observations other than those you outline in the earlier sentence.
174- the latency to approach the puzzle box was only included as the novel object test for the coyotes’ first exposure to the puzzle box correct? Should add some detail here.
177- What was their initial distance from the object? Was the object placed before coyotes were let into the enclosure from a door a certain distance away? Or was this placed while they were in the enclosure and they typically would retreat to an approximate location? Especially since the enclosures seem to be quite variable in size (0.1-1 ha) the latency could be highly variable based on enclosure size unless there’s a defined “start” time. I think this is an important methodological detail to outline for any latency measure in this study.
186- When you write that the box remained in the enclosure throughout the period, do you mean it was not removed between trials?
187- So is this a pre-test familiarization period not included in the 10 trials? It would be helpful to clarify this.
189- The peanut butter and extra food were added consistently outside of the box for every trial?
192-3 Why was the box left open on days 6 &7?
198-200 I think it’s important to point out that an object with food could elicit a different response than objects without (see Greggor et al., 2015). Your definition of boldness could include both these different responses but this may be something to clarify and potentially discuss later in relation to your results.
202- Does the dummy have a scent or is it just a visual stimulus? I’m curious if you think olfactory information could influence the response of the coyotes.
211-13 - Why combine the behaviors into categories? Perhaps due to low counts for some behaviors
220- How was time measured here? Months or days?
221- Were the outcome variables latency to approach and to touch modeled separately? Throughout this section it would be helpful to specify if outcomes are modeled separately or if any combination is happening
237- Is the proportion of time interacting and within 1 m combined or separate variables?
Throughout the Results section, please cite the tables in your supplementary materials with your model results.
268- It would be better to report the results here as no significant effects in the model of the treatment, time or interaction on the latencies.
283- I didn’t realize there was a pretrial phase for dummy, it should be described in the methods for the conspecific aggression trials. Or are you referencing a separate pilot with a coyote that was not included in this sample? It may be the terminology that is confusing, I am not sure whether the “pretrial” comprises baseline measures in October 2019 that are mentioned in the focal sampling sessions or a separate phase for these tests.
288- I think it would be helpful to describe the component loadings for the PCA, what behaviors made up the first two components of the PCA or have a supplemental figure showing the PCA results
300- Maybe add “over the course of this study” to clarify that the increase is reflecting the temporal aspect.
349-350 I would emphasize the process of habituation to the object/decreasing neophobia along with the process of learning
352-3- add Johnson-Ulrich et al., 2018 and Daniels et al., 2019 as citations for the link between persistence and problem solving
382- “approach”
426 delete period before citation
Table 1. How was directing attention defined in the investigation behavior? By the orientation of the head or body?

---

## Round 0.2 · Minor Revisions

Thank you for conducting such a thorough and responsive revision in a timely fashion. The reviewers are generally satisfied with the revision, although Reviewers 1 and 3 have a few additional comments for you to address. Notably, both ask for clarity regarding the reliability, which I agree is important. Therefore, I must ask for another minor revision before I can offer formal acceptance. I have a few small comments of my own below. Line numbers refer to the reviewing PDF line numbers
Why were these 16 coyotes selected from among ~ 90 for participation in this study? Indicating any selection criteria will help readers assess any biases in the sample.

Please rephrase lines 222-223, it reads like YOU captured the prey model because the second clause begins with “we.”

Line 285, I think “prevent” is better than “avoid” here. Also, how would the coyotes have known when a trial ended to wait to retrieve the food? Do you mean learn just to wait out the two hours from when the box was placed? It seems odd to write as if the trials are meaningful for the coyotes when it is just an interval of time specified by the researcher and the box remains after the trial ends so no end is clearly signaled.

If the coyotes used a solution once early on, and that solution was immediately locked after a single successful trial, the successful coyotes’ response to that solution likely extinguished before the next test session making it more likely that they would try a novel solution compared to if you had allowed them to continue using a successful door within a single test period. There is no perfect way to conduct these types of trials, but different methods have different implications for the likelihood of observing innovation or perseverance in the next phase, which deserves more discussion to rationalize that decision. I understand that there was little interaction with the puzzle box, so it doesn’t warrant a long discussion in the discussion section but I do think it would be good to know more about what you were thinking when you designed the procedure in such a way.

I understand what you meant on line 317 but one of the reviewers found this wording confusing. I agree with the reviewer’s suggestion to use a standard measure of reliability.

Something I missed in the initial submission but that I struggle with a bit now is the treatment of individual coyotes as the unit of analysis despite the fact that they were presented with each object in pairs (if I am correct). If you analyze the behavior as a unit, your sample size is only 8. You cannot really use a multi-level model with only eight dyads but it seems like the statistical approach does not account for the pairings in which one hunting coyote might limit the amount of time its partner could spend hunting, for example. I would imagine larger effects on latencies to approach, time spent exploring etc. Can you discuss the complications of testing in pairs and how it might impact the results?

Reviewer 1 ·

Basic reporting

No comment

Experimental design

No comment

Validity of the findings

Inter-observer reliability needs clarification. See comments below for more detail.

Additional comments

The authors have submitted a responsive, well-revised manuscript. I commend the authors for laying the groundwork for future studies investigating how changes in foraging behavior may impact other behavioral shifts in wild predators. The additional figures and tables provide clarity for the reader and support for the presented results. The revisions in the text provide the reader with additional information that allows for deeper interpretation of results, and the authors have successfully clarified many areas of the text. I have a few minor suggestions and questions discussed below. Line numbers refer to the manuscript PDF.

Line 67: This is a bit unclear. I’m assuming you mean predators incorporating more scavenged foods may alter their own behavior. Or are you referring to interactions with other predators?

Line 111: Unclear. Suggest changing to ‘and reliance on anthropogenic food may increase the prevalence of human-wildlife conflicts.’

Line 140: Would be helpful to specify what species/taxa you are referring to here.

Line 143: Change “compared” to compare.

Line 154: Cite the previous literature here.

Lines 294-296: What defined “interaction” with the puzzle box? How was this different from touching/approaching the puzzle box? Puzzle box was only a ‘novel object’ in the first presentation, correct?

Line 317: What statistical test was used to evaluate inter-observer reliability? Are you reporting 10% agreement here (or 90%)? If so, 10% is quite low agreement.

Lines 324-325: Same question as above. Reliability of simultaneous coding only 1 video is also quite limited.

Line 380: What were the sexes of the successful hunters?

Lines 412-420: I understand the necessity to report representative results here. However, the next section discussing the puzzle box suggests there were notable differences between novel object vs. puzzle box approach, which is unsurprising given the food/time associated with multiple presentations of the puzzle box. To clarify, briefly re-state that the novel object trials with the puzzle box were only the first presentation of the puzzle box.

Line 422: A clear definition of an interaction with the puzzle box is also useful here.

Line 464: Suggest adding in ‘respectively’ after “hunting trials” as persistence only applied to hunting trials.

Line 520: By “different means” do you mean innovation/problem solving? Suggest clarifying and adding supporting citation here.

Line 534: Missing ‘and’ between “reward habituated,” make “novel object” plural.

Line 587: Citations for heritability of problem solving/persistence?

Reviewer 2 ·

Basic reporting

Comments on original MS all still applicable. The revised version is of similar quality.

Experimental design

Broad comments on original MS still applicable, and revisions have addressed any of my concerns, requests for additional context, etc.

Validity of the findings

No comment.

Additional comments

The authors have extensively revised the manuscript and addressed all of my concerns. They appear to have also addressed the comprehensive comments from the other reviewers and the editor. The manuscript is much improved.

Reviewer 3 ·

Basic reporting

The authors have improved the manuscript with additional literature and background information as requested.

Experimental design

The authors have clarified their methods as requested by myself and the other reviewers.

Validity of the findings

The results are valid and the authors addressed reviewer concerns well. The information about coding reliability was a good addition, but a standard measure should be used rather than a statement about durations being within a particular percent. I would suggest reporting intraclass correlations (ICCs) as this has been a standard measure in behavioral studies.

Additional comments

The authors did an excellent job addressing all of the reviewer’s concerns including my own. I think with the previous revisions and a few more minor changes, this paper is ready for publication in Peer J.
Line by line comments based on the revised PDF:
166-169 Because you added the section about subjects, I think you can move and combine these statements with what is written in the subject section to avoid repetition
243- To help clarify that this baseline is the same as pretrial described later could rephrase to “baseline measures in a pretrial phase..”
288- Is this a reprieve or just encouraging interaction? I see how it’s a reprieve from having to work for food, but it seems like the overall purpose was to get the coyotes to interact more with the box.
Figure 1. This figure is very helpful! I think you could just point out directly where the pretrial phase was at the bottom of the figure for increased clarity.

---

## Round 0.3 · accepted · Accept

Thank you for once again being highly responsive to suggestions from the reviewers and from me. I am happy to see this innovative study published in PeerJ.